# Physical Mechanisms of Intermolecular Interactions and Cross-Space Charge Transfer in Two-Photon BDBT-TCNB Co-Crystals

**DOI:** 10.3390/nano12162757

**Published:** 2022-08-11

**Authors:** Chen Lu, Ning Li, Ying Jin, Ying Sun, Jingang Wang

**Affiliations:** 1College of Science, Liaoning Petrochemical University, Fushun 113001, China; 2Institute of Clean Energy Chemistry, Key Laboratory for Green Synthesis and Preparative Chemistry of Advanced Materials of Liaoning Province, College of Chemistry, Liaoning University, Shenyang 110036, China

**Keywords:** BDBT–TCNB co-crystals, intermolecular interactions, cross-space charge transfer, two-photon absorption

## Abstract

Co-crystal materials formed by stacking different molecules with weak interactions are a hot research topic. In this work, we theoretically investigate the intermolecular interactions and charge transfer properties of the supramolecular BDBT-TCNB co-crystal (BTC). The π-π bonds, hydrogen bonds, and S-N bonds in the BTC bind the BDBT and TCNB molecules together to form a highly ordered co-crystal and lead to the co-crystal’s excellent two-photon absorption (TPA) properties. The intermolecular interactions of the BTC are discussed in detail by the independent gradient model based on Hirshfeld partition (IGMH), atoms in molecules (AIM), electrostatic overlay diagram, and symmetry-adapted perturbation theory (SAPT) energy decomposition; it is found that there is a strong interaction force along the stacking direction. The charge transfer properties of the one-photon absorption (OPA) and TPA of the BTC were investigated by charge density difference (CDD) and transition density matrix (TDM). It is found that the dominant charge transfer mode is the cross-space charge transfer along the stacking direction. Therefore, strong intermolecular interactions will promote intermolecular cross-space charge transfer. This work is of great significance for the design of organic optoelectronic supramolecular materials.

## 1. Introduction

TPA is a third-order nonlinear optical process in which two photons excite from a medium to an intermediate state and then to an excited state. TPA has many potential applications in chemistry, life sciences, and physics. The excitation density of two photons is proportional to the square of the intensity; therefore, compared with the single photon, the two-photon excitation volume is smaller, which can improve the resolution of the microscope. This advantage has been applied to fluorescence microscopy [1,2,3]. Two-photon dyes can be used as labels to track the intracellular migration of non-fluorescent drugs [4]. Since long-wavelength lasers can minimize the damage of the light source to the catalytic substrate, TPA has a good application in the field of non-destructive photocatalysis [5,6,7].

Ordered structures formed by weak interactions between two or more molecules are called co-crystals. Co-crystals hold great promise in optics and electronics, such as light-emitting diodes and photovoltaic cells [8,9]. Co-crystals can be designed and constructed by π-π stacking structures instead of traditional π-conjugated structures. The properties of co-crystals are determined by their structure, and different packing structures lead to different intermolecular interactions in the system. Electron delocalization or charge interactions in π-π stacking systems have important effects on the optoelectronic properties [10], absorption, emission [11,12], stability [13], and photoluminescence quantum efficiency of co-crystals. Yang et al. demonstrated that the electrostatic and dispersive interactions of conjugated carbon materials relative to other materials are critical to the performance of photovoltaic devices [14,15,16].

The BTC is composed of benzo [b] naphtho [1,2-d] thiophene (BDBT) and 1,2,4,5-tetracyanobenzene (TCNB) stacked on each other, which have excellent TPA properties [17]. In this work, we chose BTC to study intermolecular interactions and two-photon transition properties. Conjugated structures are widely used in optoelectronic materials due to their high charge transfer ability. Strong intermolecular interactions will be more favorable for charge transfer, and this work also proves that charge transfer is more inclined to transfer along the direction of strong interactions. This provides the necessary theoretical basis for designing other optoelectronic materials.

## 2. Materials and Methods

All structures in this work were derived from the crystallographic information file (CIF) of the BTC. Electron excitation calculations were performed by the Gaussian program [18] combined with the density functional theory (DFT) [19], CAM-B3LYP functional [20], and the 6–31 g(d) basis set [21]. The energy decomposition was calculated by the psi4 program [22], combining sSAPT0 levels and jun-cc-pVDZ. The interaction energy obtained from this combination has been shown to be very close to the experimentally measured results [23]. All wave function analyses (IGMH, AIM, electrostatic potential overlay, CDD, and TDM) were carried out by the Multiwfn program [24]. The TPA spectrum was calculated based on a script we wrote ourselves [25]. All 3D structure diagrams in this work were drawn by the VMD program [26].

IGMH [27] is a method that can visualize the interactions in chemical systems in a graphical way. It is defined as:(1)$$\delta g\left(r\right)=g^{IGM}\left(r\right)-g\left(r\right)$$
(2)$$g\left(r\right)=\left|\sum_{i}\nabla \rho_{i}\left(r\right)\right|$$
(3)$$g^{IGM}\left(r\right)=\sum_{i}\left|\nabla \rho_{i}\left(r\right)\right|$$
where $g^{IMG}$ is the sum of the absolute values of the electron density gradients; g is the sum of the electron density gradients; $\rho_{i}$ represents the electron density of the i atom; r is a coordinate vector; and $\delta g^{\mathrm{inter}}$ and $\delta g^{\mathrm{intra}}$ are defined to reflect inter- and intra-fragment interactions, respectively.
(4)$$\delta g^{\mathrm{inter}}\left(r\right)=g^{\mathrm{IGM},\mathrm{inter}}\left(r\right)-g^{\mathrm{inter}}\left(r\right)$$
(5)$$g^{\mathrm{inter}}\left(r\right)=\left|\sum_{A}\sum_{i\in A}\nabla \rho_{i}\left(r\right)\right|$$
(6)$$g^{\mathrm{IGM},\mathrm{inter}}\left(r\right)=\sum_{A}\left|\sum_{i\in A}\nabla \rho_{i}\left(r\right)\right|$$
(7)$$\delta g^{intra}\left(r\right)=\delta g\left(r\right)-\delta g^{inter}\left(r\right)$$

The two-photon molar absorptivity [25] is defined as:(8)$$\begin{array}{l} \delta_{tp}=8\sum_{\begin{array}{l} j\neq g\\ j\neq f \end{array}}\frac{{\left|\langle f|\mu |j\rangle \right|}^{2}{\left|\langle j|\mu |g\rangle \right|}^{2}}{{\left(\omega_{j}-\omega_{f}/2\right)}^{2}+\Gamma_{f}^{2}}\left(1+2\cos^{2}\theta_{j}\right)\\ +8\frac{{\left|\Delta \mu_{fg}\right|}^{2}{\left|\langle f|\mu |g\rangle \right|}^{2}}{{\left(\omega_{f}/2\right)}^{2}+\Gamma_{f}^{2}}\left(1+2\cos^{2}\phi \right) \end{array}$$
where $|g\rangle$, $|j\rangle$, and $|f\rangle$ are the wave functions of the ground state, intermediate state, and final state during the TPA process, respectively; $\langle j|\mu |g\rangle$ and $\langle f|\mu |j\rangle$ are the transition dipole moments from the ground state to the intermediate state and the intermediate state to the final state, respectively; $\theta_{j}$ is the angle between the two transition dipole moments; $\Delta \mu_{fg}$ is the difference in permanent dipole moments between the ground state and the final state; ϕ is the angle between $\Delta \mu_{fg}$ and $\langle f|\mu |g\rangle$; $\omega_{j}$ and $\omega_{f}$ are the energies of the intermediate state and the final state, respectively; and $\Gamma_{f}$ is the lifetime of the ground state.

## 3. Results and Discussion

BTC (Figure 1) is an excellent two-photon material. The BTC can well preserve the two-photon properties of the donor BDBT, and there are obvious charge transfer interactions in the BTC. Intermolecular interactions are crucial for the optical properties of co-crystals. In this section, we firstly study the intermolecular interactions of the BTC in detail and then visualize the OPA and TPA charge transfer of the co-crystals.

### 3.1. Intermolecular Interactions in the BTC

Intermolecular interactions are crucial for the charge transfer properties of the co-crystals. In this section, the interactions between dimers of four different configurations of BDBT and TCNB in the BTC were investigated by IGMH, AIM, and SAPT energy decomposition.

IGMH is a method that can visualize the interactions in chemical systems in a graphical way. Dimer 1 and dimer 3 are composed of BDBT and TCNB stacked on each other, and a large-scale green interaction isosurface is formed between the molecules (Figure 2a,c); this indicates that there is a significant π-π stacking interaction between BDBT and TCNB. Grimme had effectively demonstrated that the dispersive interactions of π electrons in the packing direction in unsaturated molecules is the essence of the π-π stacking effect [28]. Therefore, there are strong dispersive interactions in dimer 1 and dimer 3. Dimers 2 and 4 are connected laterally by N-H and S-N bonds to BDBT and TCNB (Figure 2b,d). The intermolecular green isosurfaces also represent van der Waals interactions. Electrostatic interactions are also the main source of intermolecular attraction. The electrostatic potential overlay map can clearly show the region of intermolecular electrostatic interaction [29]. The larger the overlap of the electrostatic potentials between the two molecules in the opposite sign, the stronger the electrostatic attraction. It can be seen that in dimer 1 and dimer 3 (Figure 3a,c), the areas with opposite signs of electrostatic potential overlap greatly, while dimer 2 and dimer 4 only have a small overlap (Figure 3b,d); this suggests that the electrostatic interactions in dimers 1 and 3 also contribute significantly to the mutual attraction of the molecules. The electrostatic interaction in dimer 4 is stronger than in dimer 2 because of the large overlap of electrostatic potentials in dimer 4 (Figure 3b,d).

AIM [30,31,32] is an important wave function analysis method and one of the most popular methods used for analyzing chemical bonds. AIM mainly obtains the relevant properties of chemical bonds by analyzing bond critical points (BCPs). When studying the interactions between atoms, it is natural to think of starting from the electronic structure characteristics of the interactions between atoms. In AIM theory, BCPs are considered to be the most representative points in the interatomic interaction region. So, the properties of BCPs can be used to study the properties of the corresponding chemical bonds, including the strengths and properties. The orange lines between the molecules in Figure 2 are the interaction paths between atoms, and the red spheres in the paths are the BCPs. The electron density and the energy density (Figure 4a) at all BCPs are very low (electron density < 0.01 a.u. and energy density < 0.002 a.u.). Both the electron density and the energy density reflect the very typical characteristics of non-covalent interactions. At the same time, more real-space functions of BCPs are calculated (Appendix A), which will be more helpful in understanding the intermolecular interactions in the BTC.

Finally, we performed energy decomposition calculations for the intermolecular interactions of the four dimers to better understand the nature of the interactions. SAPT is a very popular method used to decompose weak interactions [33,34,35,36]. Energy decomposition can decompose the inter-fragment interaction energy into different physical components (dispersive, electrostatic, induced, and exchange interactions) and then gain a deeper understanding of the nature of the interaction from the energy perspective. It can be seen from Figure 4b that the total interaction energy of dimer 1 and dimer 3 is significantly stronger than that of dimer 2 and dimer 4; this suggests that dimer 1 and dimer 3 are more stable. The energy decomposition results show that, among the four configurations, exchange interactions play a repulsive role, and electrostatic, dispersive, and induced interactions play an attractive role. Among them, dispersive interactions play a major contribution in the binding of molecules. However, electrostatic interactions are also not negligible. Among them, dimer 1 and dimer 3 have stronger electrostatic interactions, and dimer 2 has the smallest electrostatic interactions; this is consistent with the conclusion obtained from the electrostatic potential overlay diagrams. The numerical values of the energy decomposition results are shown in Appendix A. In all configurations, dispersive interactions contributed to more than 50% of the total attraction, dominating the intermolecular binding.

### 3.2. Cross-Space Charge Transfer in the BTC

#### 3.2.1. The OPA and TPA Spectrum

Figure 5a shows the OPA spectrum of the BTC. After comparing this with the spectrum measured by the experiment [17], it is found that the calculated spectrum is accurate. There is an absorption peak mainly contributed by S_2_ (432.99 nm), S_4_ (411.59 nm), S_6_ (405.95 nm), and S_7_ (400.93 nm) in the range of 350–450 nm. Figure 5b shows the TPA spectrum of the BTC. There is a strong absorption peak and a weak absorption peak in the range of 700–900 nm; the strong absorption peak is contributed by S_5_ (813.10 nm), S_7_ (802.98 nm), and S_8_ (802.93 nm), and the weak absorption peak is mainly contributed by S_10_ (754.17 nm). The two absorption peaks were both determined by one-step transition and two-step transition.

#### 3.2.2. Electronic Transition Properties of OPA

CDD and TDM are currently popular methods used to visualize the electron transfer process [37,38,39,40], which can clearly reflect where the electrons come from and where they go. The atomic number corresponding to the TDM is shown in Appendix A. There are four main excited states in the OPA spectrum, namely S_2_, S_4_, S_6_, and S_7_. In this section, we qualitatively analyze the electronic transition properties of each excited state by CDD and TDM. At the same time, we also quantitatively analyze the excited states by the transition indexes. Overall, the four one-photon excited states are all electron transfer excited states (Figure 6), and the electrons are all transferred from BDBT (donor) to TCNB (acceptor). The isosurfaces of the electrons and holes of S_2_ converge on the four molecules in the middle of the system, and the electrons are transferred from BDBT to TCNB (Figure 6a,b). The centroid distance (D) of the electrons and holes in S_2_ is only 0.043 Å (Table 1) because the electron and hole isosurfaces have the same center of symmetry; this does not mean that the electrons and holes of S_2_ are not separated. The electron transfer process of S_4_ is obviously different from that of S_2_. The electron and hole isosurfaces of S_4_ are located on the four molecules on both sides of the system, and the electrons are also transferred from BDBT to TCNB (Figure 6c,d); the electron and hole isosurfaces on the right side of the system are significantly larger than those on the left side. The average distribution breadth (H) of the electrons and holes in S_4_ reaches 12.196 Å (Table 1), which is caused by the distribution of electrons and holes on both sides of the system. The electron transfer process of S_6_ is very similar to that of S_4_ (Figure 6e,f), and the isosurfaces of the electrons and holes are located on both sides of the system; the H of S_6_ is 11.674 Å (Table 1). S_7_ has only one set of electron–hole isosurfaces (Figure 6g,h), which is why the D of S_7_ is significantly larger than the other three excited states. The electron–hole overlap index (Sr), hole delocalization index (HDI), and electron delocalization index (EDI) of the four excited states are all very close (Table 1); this shows that the degree of overlap and delocalization of the electrons and holes in the four excited states is very close. The electron–hole separation index (t) can also measure the separation degree of the electrons and holes (Table 1). The electrons and holes are sufficiently separated when t > 0, and vice versa when t < 0. The electron–hole densities of S_2_, S_4_, and S_6_ are all well separated, but t is negative. This is because the three excited states have two groups of electron–hole isosurfaces; as a result, t cannot effectively reflect the degree of separation of the electrons and holes.

#### 3.2.3. Electronic Transition Properties of TPA

Two-photon materials are widely used in 3D optical imaging, lithographic micromachining, and optical data storage. In this section, we investigate the electronic transition mechanism of the excited states with a large cross-section in the BTC. S_5_, S_7_, S_8_, and S_10_ have larger two-photon cross-sections in the TPA spectrum of the BTC (Figure 5b). We visualized the first and second transitions in the two-photon transition processes by CDDs and TDMs. S_5_ has one transition channel (S_0_→S_4_→S_5_). S_4_ in OPA is an excited state with strong oscillator strength, so S_4_ is likely to become an intermediate state in the TPA processes (S_2_, S_6_, and S_7_ may also become intermediate states). The electronic transition process of S_0_→S_4_ is the same as that of S_4_ in OPA (Figure 7c,d). The electrons and holes isosurfaces of S_4_→S_5_ are located at the four BDBTs at the corners of the system (Figure 7a,b), and electrons are transferred from one BDBT across the TCNB to the other BDBT; this reflects the strong charge transfer ability of the BTC. S_8_ has the strongest two-photon cross-section in the TPA spectrum, with only one transition channel, and the intermediate state is S_7_ (S_0_→S_7_→S_8_). The electronic transition process of S_0_→S_7_ is the same as that of S_7_ in OPA (Figure 7g,h). The electrons transfer of S_7_→S_8_ is upward as a whole (Figure 7e,f), which is reflected as the blue isosurfaces representing the holes are below the red isosurfaces representing the electrons. The wavelength of S_7_ is very close to that of S_8_, but the two-photon cross-section is lower than that of S_8_. S_7_ has two transition channels (Figure 8), S_0_→S_2_→S_7_ and S_0_→S_6_→S_7_, respectively. The first-step transition of the two channels is the same as the transition of the corresponding excited state in the OPA. The second transition (S_2_→S_7_) of S_0_→S_2_→S_7_ is the charge transfer excitation of six molecules located in the middle of the system (Figure 8a,b). The electrons are transferred from the upper and lower sides to the middle, which is reflected in the red isosurfaces representing the electrons located in the middle of the system. The electron and hole isosurfaces of the second transition (S_6_→S_7_) of S_0_→S_6_→S_7_ are mainly located in the four molecules on the left side of the system (Figure 8e,f), and the transfer direction of electrons is downward. S_10_ dominates a weak absorption peak; it has two transition channels, S_0_→S_6_→S_10_ and S_0_→S_9_→S_10_, respectively. The second-step transition (S_6_→S_10_) of S_0_→S_6_→S_10_ is a charge transfer excitation located on both sides of the system (Figure 9a,b), and the electrons are transferred downward as a whole. The electronic excitation process of S_0_→S_9_→S_10_ (Figure 9e–h) is similar to that of S_0_→S_6_→S_10_, which can be seen from the CDDs and TDMs. From the size of the isosurface, it can be seen that the electron transfer intensity of the second channel is stronger than that of the first channel. Finally, we give the transition dipole moments of the individual transitions during the two-photon excitation (Table 2). The numerical values of the transition dipole moments are closely related to the strength of the two-photon cross-section.

In summary, we find that electrons are mostly transferred along the π-π stacking direction because the intermolecular interactions along the stacking direction are stronger. This will provide guidance for designing materials with excellent charge transfer capabilities.

## 4. Conclusions

In this work, we investigated the intermolecular interactions and optical properties of the BTC with a packing structure using first-principles calculations and wave function analysis. We discussed the two main driving forces for molecular binding, dispersive and electrostatic forces, through the IGMH and electrostatic overlay diagrams and give regions for the dispersive and electrostatic interactions. A series of real-space functions such as electron density and energy density at the intermolecular interactions were calculated by AIM. The energy dissociation of dimers with different binding modes was calculated using the SAPT theory, which helps us understand the nature of intermolecular interactions. BDBT and TCNB formed the BTC through π-π stacking, and abundant intermolecular interactions such as ππ bonds, hydrogen bonds, and N-S bonds were formed in the co-crystals. Strong intermolecular interactions will promote charge transfer between molecules. Charge transfer in the co-crystals is mainly achieved by cross-space charge transfer. We performed a visual study of the electron transfer in the OPA and TPA of the BTC by CDDs and TDMs and clarified the charge transfer mechanism in the co-crystals. We explained why the excited state with large oscillator strength in single-photon absorption is more likely to be the intermediate state in the TPA. Electrons tend to transfer in the direction of strong interactions. This work will provide the necessary theoretical basis for designing luminescent materials with a large two-photon cross-section.

## Figures and Tables

**Figure 1 nanomaterials-12-02757-f001:**
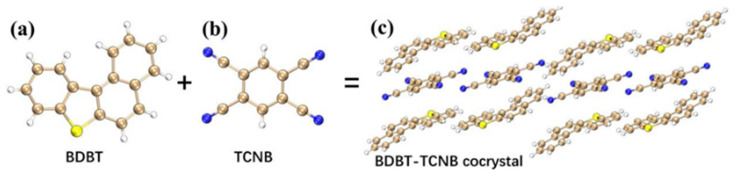
The structural diagrams of (**a**) BDBT, (**b**) TCNB, and (**c**) BDBT–TCNB co-crystals. Gold, blue, yellow, and white spheres represent carbon, nitrogen, sulfur, and hydrogen atoms, respectively.

**Figure 2 nanomaterials-12-02757-f002:**
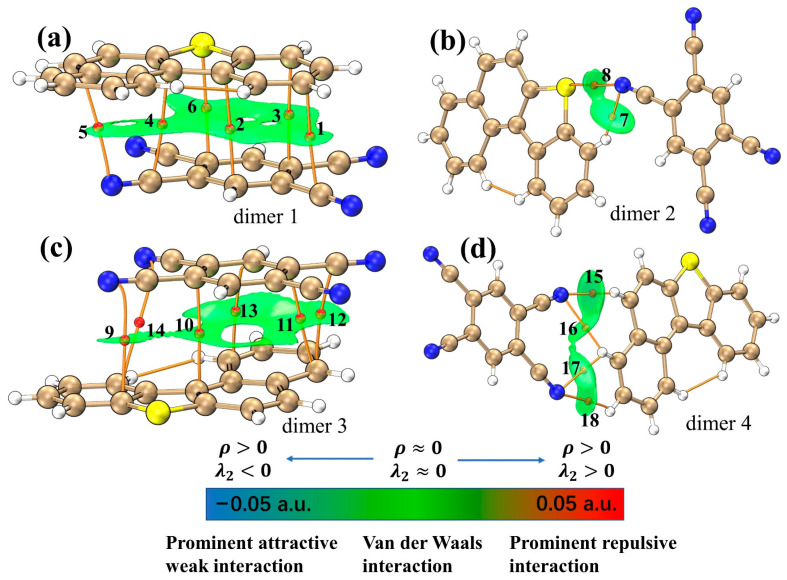
IGMH diagrams for (**a**) dimer 1, (**b**) dimer 2, (**c**) dimer 3, and (**d**) dimer 4. The orange lines are interaction paths, and the red spheres are bond critical points.

**Figure 3 nanomaterials-12-02757-f003:**
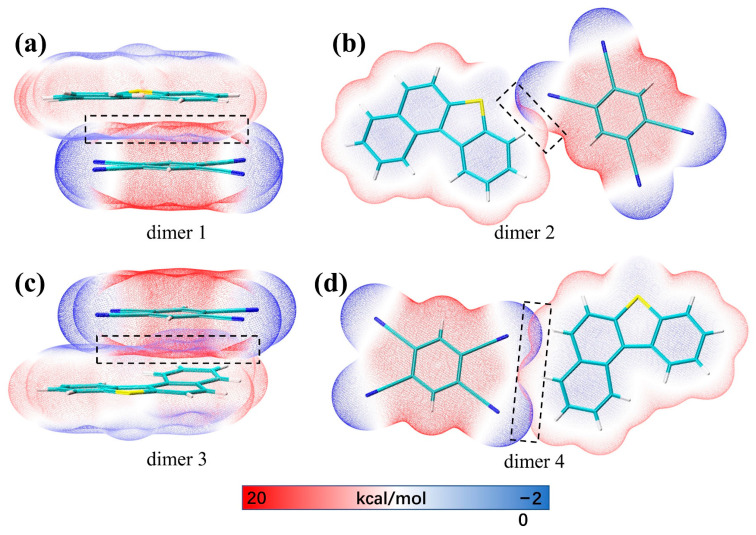
Electrostatic potential overlay diagrams for (**a**) dimer 1, (**b**) dimer 2, (**c**) dimer 3, and (**d**) dimer 4.

**Figure 4 nanomaterials-12-02757-f004:**
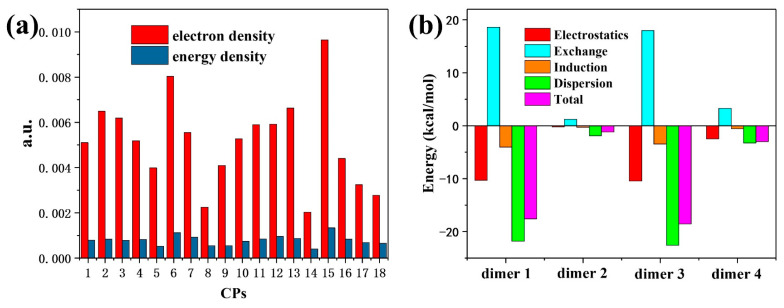
(**a**) The electron density and energy density at the BCPs. (**b**) The total interaction energy of dimers and their components.

**Figure 5 nanomaterials-12-02757-f005:**
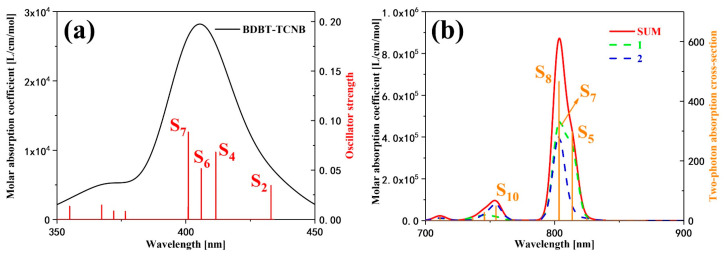
(**a**) The one-photon absorption spectrum and (**b**) the two-photon absorption spectrum of the BTC.

**Figure 6 nanomaterials-12-02757-f006:**
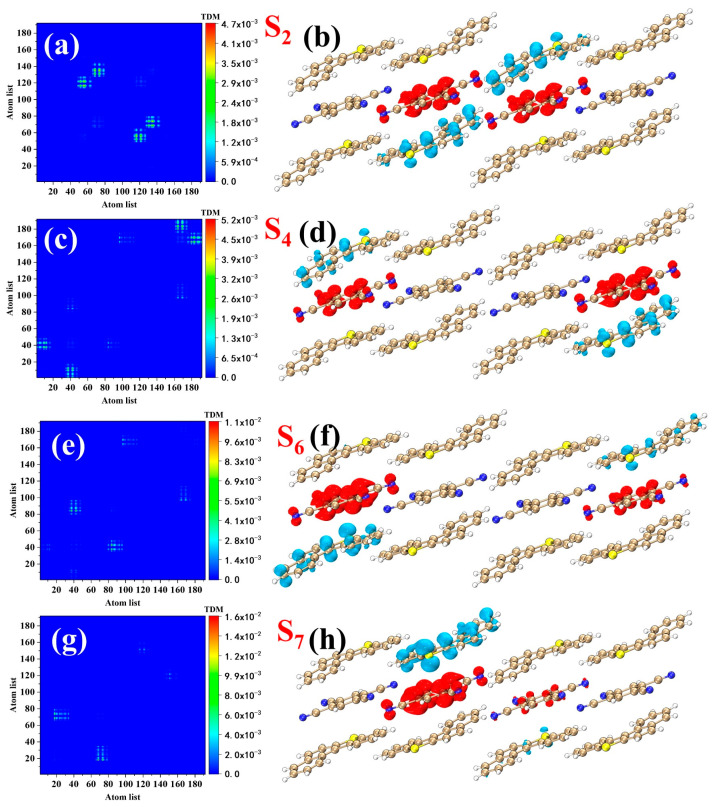
CDDs and TDMs of (**a**,**b**) S_2_, (**c**,**d**) S_4_, (**e**,**f**) S_6_, and (**g**,**h**) S_7_ in one-photon absorption. The red and blue isosurfaces represent the electrons and holes, respectively.

**Figure 7 nanomaterials-12-02757-f007:**
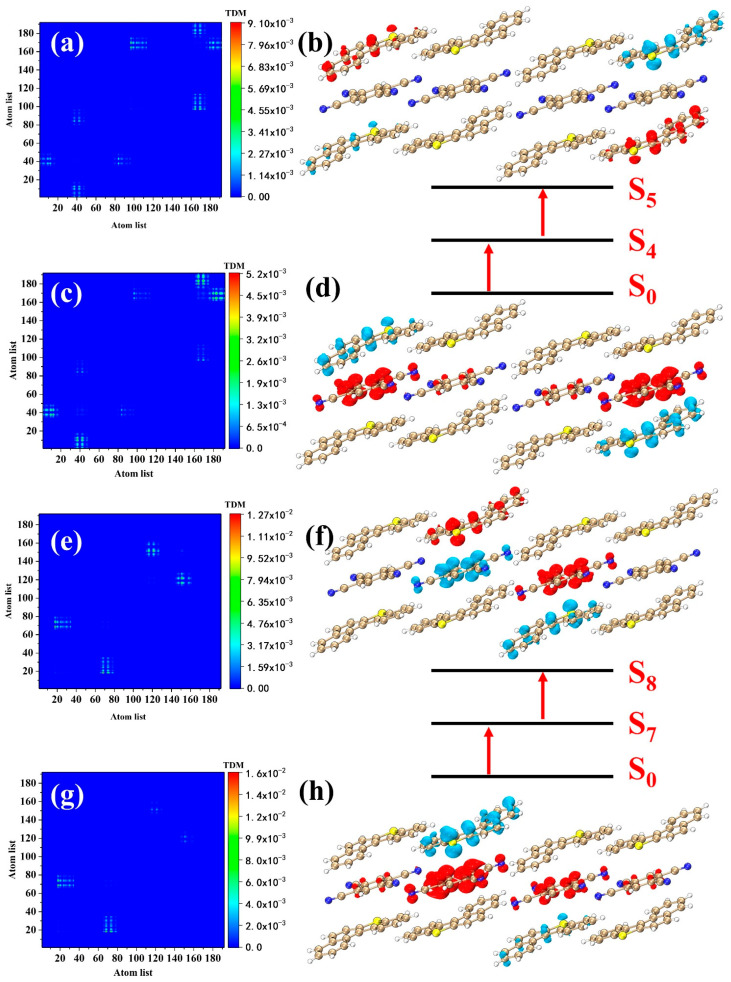
CDDs and TDMs of (**a**,**b**) S_4_→S_5_, (**c**,**d**) S_0_→S_4_, (**e**,**f**) S_7_→S_8_, and (**g**,**h**) S_0_→S_7_ in two-photon absorption. The red and blue isosurfaces represent the electrons and holes, respectively.

**Figure 8 nanomaterials-12-02757-f008:**
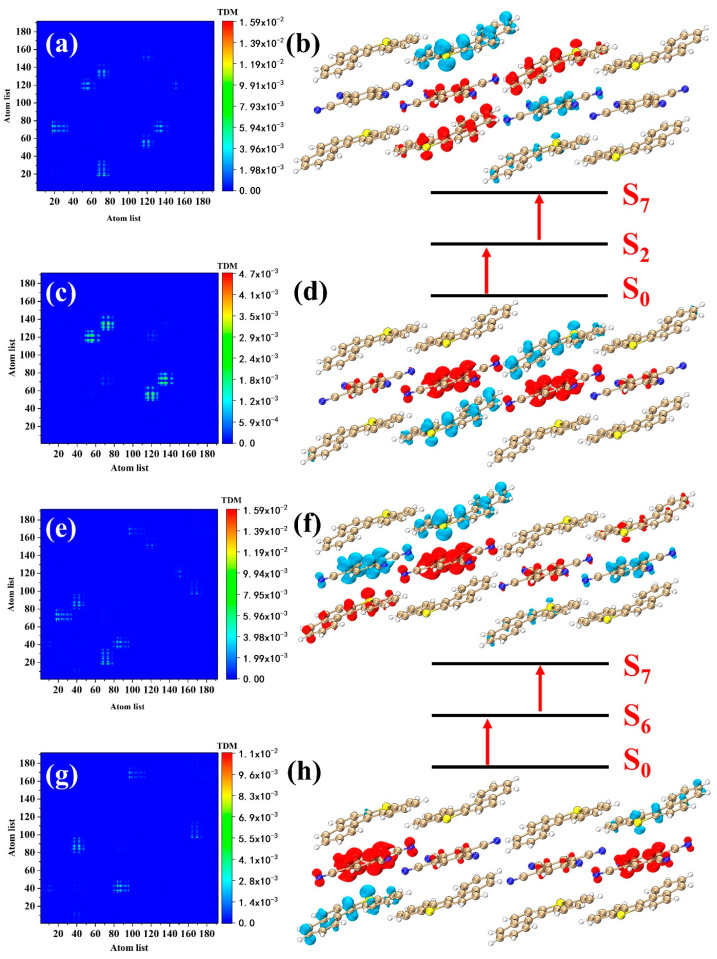
CDDs and TDMs of (**a**,**b**) S_2_→S_7_, (**c**,**d**) S_0_→S_2_, (**e**,**f**) S_6_→S_7_, and (**g**,**h**) S_0_→S_6_ in two-photon absorption. The red and blue isosurfaces represent the electrons and holes, respectively.

**Figure 9 nanomaterials-12-02757-f009:**
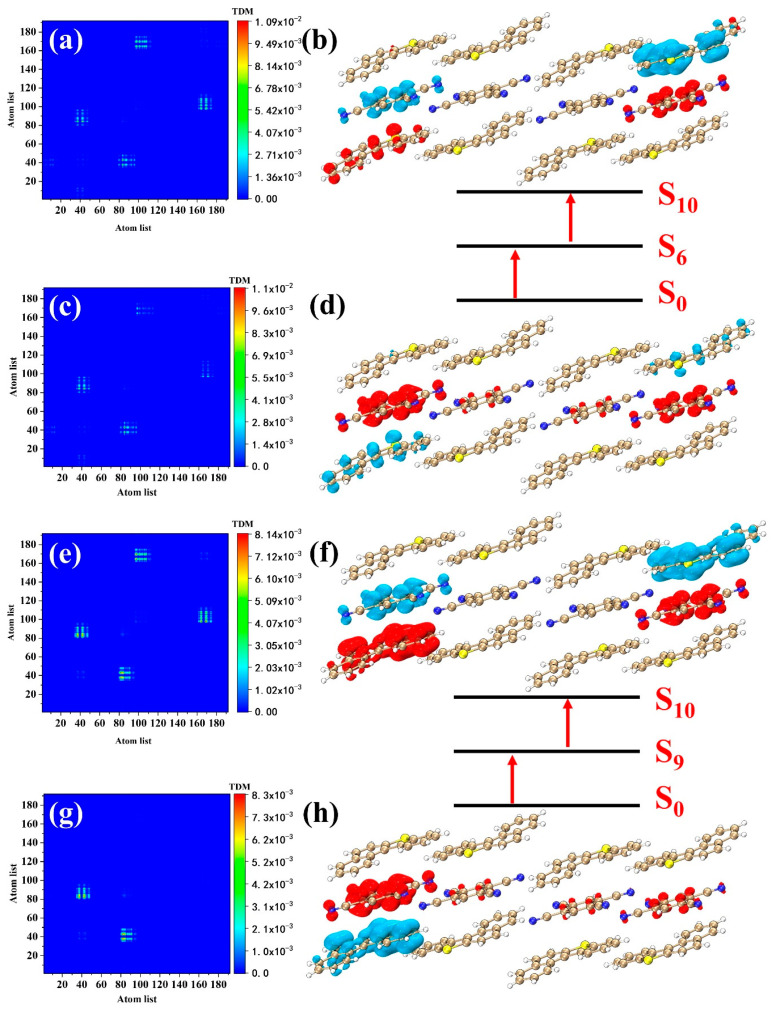
CDDs and TDMs of (**a**,**b**) S_6_→S_10_, (**c**,**d**) S_0_→S_6_, (**e**,**f**) S_9_→S_10_, and (**g**,**h**) S_0_→S_9_ in two-photon absorption. The red and blue isosurfaces represent the electrons and holes, respectively.

**Table 1 nanomaterials-12-02757-t001:** Transition indices of S_2_, S_4_, S_6_, and S_7_ in one-photon absorption.

	D (Å)	Sr	H (Å)	t (Å)	E (eV)	HDI	EDI
S_2_	0.043	0.202	5.700	−2.597	2.863	4.46	5.51
S_4_	0.475	0.219	12.196	−2.408	3.012	3.86	5.57
S_6_	1.104	0.264	11.674	−3.284	3.054	4.18	5.89
S_7_	2.641	0.239	4.899	0.193	3.092	5.05	6.56

**Table 2 nanomaterials-12-02757-t002:** Transition dipole moments of the first-step transition and the second-step transition of each excited state in two-photon absorption.

TPA States	Process	Transition Dipole Moment
S_5_	$\langle \phi_{S_{0}}|\mu |\phi_{S_{4}}\rangle \to \langle \phi_{S_{4}}|\mu |\phi_{S_{5}}\rangle$	1.135→24.263
S_7_	$\langle \phi_{S_{0}}|\mu |\phi_{S_{2}}\rangle \to \langle \phi_{S_{2}}|\mu |\phi_{S_{7}}\rangle$	0.596→0.143
	$\langle \phi_{S_{0}}|\mu |\phi_{S_{6}}\rangle \to \langle \phi_{S_{6}}|\mu |\phi_{S_{7}}\rangle$	0.786→0.054
S_8_	$\langle \phi_{S_{0}}|\mu |\phi_{S_{7}}\rangle \to \langle \phi_{S_{7}}|\mu |\phi_{S_{8}}\rangle$	1.422→28.729
S_10_	$\langle \phi_{S_{0}}|\mu |\phi_{S_{6}}\rangle \to \langle \phi_{S_{6}}|\mu |\phi_{S_{10}}\rangle$	0.786→0.291
	$\langle \phi_{S_{0}}|\mu |\phi_{S_{9}}\rangle \to \langle \phi_{S_{9}}|\mu |\phi_{S_{10}}\rangle$	0.079→3.477

## Data Availability

Data can be available upon request from the authors.

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
