# Peer review of "Physical Mechanisms of Intermolecular Interactions and Cross-Space Charge Transfer in Two-Photon BDBT-TCNB Co-Crystals"

_nanomaterials, 2022, doi:10.3390/nano12162757_

Round 1

Reviewer 1 Report

This is an interesting and well presented modeling paper. I only wonder how precisely the results relate to experimentally observed data. Could some more discussion about the relevance to experimentally observed results be added, please?

Reviewer 2 Report

The article is devoted to a detailed analysis of the mechanism of intermolecular interactions in the co-crystal material BDBT-TCNB formed by stacking weakly interacting donor and acceptor molecules. The topic of the work is relevant, the choice of the object of research is justified. The used calculation approaches correspond to the aims of the work.

Small remarks: 1. Reference should be made to well-known publications where formula (8) is substantiated.

2. It would make sense to discuss in more detail the possible contribution of different (1-4) types of dimers to the experimentally observed in [17] photophysical properties of BDBT-TCNB co-crystals.

The manuscript can be accepted in this form and does not require revision
